# Herpes Zoster Vaccination Rates in Hematological and Oncological Patients—Stock Taking 2 Years after Market Approval

**DOI:** 10.3390/healthcare10081524

**Published:** 2022-08-12

**Authors:** Til Ramón Kiderlen, Katrin Trostdorf, Nicola Delmastro, Arne Salomon, Maike de Wit, Mark Reinwald

**Affiliations:** 1Department of Hematology, Oncology and Palliative Care, DRK Clinics Berlin Koepenick, 12559 Berlin, Germany; 2Faculty of Health Sciences Brandenburg, Brandenburg Medical School Theodor Fontane, 14770 Brandenburg an der Havel, Germany; 3Department of Hematology, Oncology and Palliative Care, Vivantes Hospital Neukoelln, 12351 Berlin, Germany; 4Department of Cardiology, Angiology, Nephrology, and Intensive Care Medicine, Vivantes Hospital Neukoelln, 12351 Berlin, Germany; 5Department of Pulmonology and Infectious Diseases, Vivantes Hospital Neukoelln, 12351 Berlin, Germany; 6Department of Hematology and Oncology, Brandenburg Medical School Theodor Fontane, 14770 Brandenburg an der Havel, Germany

**Keywords:** Herpes zoster, vaccination, cancer patients, oncology

## Abstract

Background: Vaccinations have the potential to significantly lower the burden of disease for many major infections in the high-risk population of hematological and oncological patients. In this regard Shingrix^®^, an inactivated *Varicella Zoster Virus* vaccine, received market approval in the European Union in March 2018, after prior US approval in October 2017, and recommendations specifically state immunocompromised, including oncological, patients. As vaccination rates are considered to be poor in oncological patients, determining the current vaccination rates for Shingrix^®^ two years after market approval is important in defining the need for intervention to bring this potentially high-impact vaccine to the patients. Methods: We analyzed data of the EVO Study to provide data for Herpes zoster vaccination rates in oncological patients. The EVO Study was an interventional study evaluating the potential of increasing vaccination rates of specified must-have vaccinations by an instructional card in the oncological setting. Numbers presented in this publication merged baseline data and follow-up data of the control group; hence data not affected by the intervention. Results: Data of 370 patients were analyzed; 21.1% with hematological malignancies and 78.9% with solid cancer. Only 3.0% were vaccinated with Shingrix^®^. Patients with hematological malignancy were more likely to be vaccinated than those with solid cancer (7.7 vs. 1.7%). Conclusion: Despite clear recommendations and a pressing need in the high-risk population of hematological and oncological patients, the vast majority of patients are still left without vaccine protection against Herpes zoster by Shingrix^®^.

## 1. Introduction

Herpes zoster (HZ) is the result of reactivation of the latent *Varicella zoster virus* (VZV) dormant in nerve ganglia, presenting usually with dermatomal manifestations such as vesicular rash and painful neuritis, potentially causing chronic neuropathic pain and leading to high morbidity. More than 90% of adults carry VZV, following infection during childhood, and are therefore at risk of reactivation [1]. A compromised immune system is the main reason for HZ [2], including hematological and oncological conditions [3]. While incidence for HZ is 3.2 per 1000 person years in the overall population [4], this number rises to about 12 in patients with solid tumors and 31 in patients with hematological diseases [3]. Therefore, in hematological high-risk patients a HZ prophylaxis is recommended as it has been shown to provide a positive impact on mortality [5]. Specific antineoplastic or immunosuppressive medication further increases risk of VZV reactivation [6]. Consequently, prevention of HZ is of utmost importance for cancer patients. Until recently, the only vaccine available was the live attenuated Zostavax^®^ (Merck&Co., Kenilworth, USA), which is contraindicated in immunocompromised patients. In October 2017 the subunit vaccine Shingrix^®^ (GlaxoSmithKline Biologicals, London, UK), based on VZV glycoprotein E (gE), was approved for individuals 50 years and older in the US and in March 2018 in the EU. In December 2018 the German Standing Committee on Vaccination (Ständige Impfkommission, STIKO) recommended Shingrix^®^ for all individuals age 60 years and above and for individuals age 50 years and above with disease-related or induced immunosuppression or other illness-related conditions, including hematological and oncological patients [7]. Even though for the US (July 2021) and the EU (August 2021) drug authorities already approved usage for patient age 18 years and older with elevated risk for HZ, the STIKO recommendation for Germany stands until today [8].

Vaccination coverage in oncological patients is generally considered to be low [9], but real-life data of actual vaccination rates in hematological and oncological patients are scarce. In order to provide first data for HZ we analyzed the results of the *Easy Vaccination in Oncology (EVO)* Study, conducted the second year after market approval, to provide data for interventional studies and planning of health care interventions.

## 2. Material and Methods

Data presented is derived from the EVO Study, a before-after study evaluating the effect of an intervention intending to increase vaccination rates in the population of oncological patients for a predefined subset of recommended vaccinations [10]. The study took place between 2 January 2020 and 31 January 2021 in outpatient clinics in Berlin, Germany. Oncological patients, including those with hematological malignancies, under antineoplastic treatment, were eligible. In order to provide data without influence of the EVO intervention, only patients of the baseline survey and of the follow-up survey of the control group (not the intervention group) were included in the analysis presented here. Evaluation of vaccination status was done by assessing vaccination cards. Status was documented as vaccinated according to STIKO recommendations; this included having received only one vaccination with Shingrix^®^ less than 6 months ago as first shot, assuming upcoming completion with the second shot. In case no vaccination card existed, vaccinations were assessed as “not vaccinated”, according to international recommendations.

Secondary diagnosis (SD) was documented and grouped to analyze additional rationales for vaccination. Grouping was done as follows: cardiac SD (Hypertension, Arrhythmia, post myocardial infarction, coronary heart disease, cardiac heart failure, cardiomyopathy), pulmonary SD (chronic obstructive pulmonary disease/COPD, bronchial asthma, interstitial lung disease), nephrological SD (renal failure), vascular SD (thrombosis, embolism, thrombocytopathy, disorders of coagulation), gastrointestinal SD (colitis, chronic hepatitis, pancreatitis, chronic cholecystitis), hepatic SD (hepatitis), endocrine SD (diabetes mellitus, hyper- or hypothyroidism) and autoimmune SD (rheumatic diseases, arthritis).

The study was registered at the German Registry for Clinical Studies (DRKS) under number DRKS00020118. The study was approved by the ethics committee of the Brandenburg Medical school Theodor Fontane.

## 3. Results

A total of 370 patients (71.3%) of the EVO study population (519) were included in the analysis. Sex was equally distributed (50.8% female vs. 49.2% male). Mean age was 68 years, with 94.1% being 50 years and older. Twenty-one percent of patients were treated for hematological malignancy and 78.9% for solid cancer.

Only 3.0% were vaccinated against HZ with Shingrix^®^, with no difference between women and men (3.2 vs. 2.7%). Furthermore, for 0.5% a valid vaccination with Zostavax^®^ was documented.

Patients with hematological malignancies were more frequently vaccinated with Shingrix^®^ than patients with solid tumors (7.7% vs. 1.7%; see Table 1). In the subgroup of patients with hematological malignancies, the highest rate was observed for patients with multiple myeloma (11.6%). Of those 39 patients receiving therapeutic agents with a known higher risk of HZ reactivation (immunomodulators, JAK-2-Inhibitors, proteasome inhibitors, long-term chronic steroid) all were treated for Multiple Myeloma. Only two of these patients (5.1%) were vaccinated with Shingrix^®^, while one patient was vaccinated with Zostavax^®^.

When analyzing secondary diagnosis (SD) as a potential additional indication for vaccination with Shingrix^®^ vaccination rates were lower in patients with documented SD (see Table 2). Of 17 patients with chronic autoimmune diseases as an additional condition, none were vaccinated against VZV.

## 4. Discussion

Compared to the general population, rates of HZ reactivation are considerably higher in hematological and oncological patients with a standardized incidence rate of 4.8 and 1.9, respectively [3]. In this regard, the inactivated *Varicella zoster* vaccine Shingrix^®^ has been shown to induce strong immune responses in hematological and oncological patients [11,12]. Although the impressive efficacy of about 97% for the general elderly population [13] could not be replicated in real-world data for immunocompromised patients (64%) [14], vaccination has the potential to significantly reduce disease burden and morbidity in this high-risk population. In this regard, real-life documented vaccination rates we observed in our study population are disappointingly low. Additionally, considering approximately only two years of vaccine availability, a rate of 3% clearly indicates a lack of awareness. Moreso as the Joint Federal Committee (Gemeinsamer Bundesausschuss) of Germany, following the recommendation of the STIKO, established Shingrix^®^ as a standard vaccination for immunocompromised patients in March 2019. Accordingly, all hematological and oncological patients in Germany of age 50+ are legally entitled to be vaccinated, and involved physicians are obliged to offer this to the patient. In this light a vaccination rate close to none seems unacceptable in the year 2020, even more so as we were able to increased rates to 18% after only 3 months facilitating an offer to be vaccinated by a simple intervention, showing the potential to rapidly booster numbers [10].

The moderately higher (7.7%) vaccination rates in hematological patients might indicate more awareness regarding substantially elevated HZ incidence rates in this patient group [3]. Interestingly, with 11.6% we found the highest vaccination rates for patients with multiple myeloma. Multiple myeloma is with an incidence rate of 56/1000 person years known to be most at risk for HZ reactivation due to illness, but also treatment related immunocompromising conditions [3]. As most myeloma patients usually receive antineoplastic medication known to significantly elevate the risk for HZ, prophylactic application of acyclovir is standard of care [5], potentially triggering other preventive measures like vaccination. Nonetheless, of those patients being treated with these antineoplastic drugs, only 5% were vaccinated. Even though efficacy of vaccination during treatment with these drugs may be reduced [6], weighing the potential benefit against the limited risks still justifies vaccine administration according to recommendations.

As STIKO recommendations cover immunosuppression as well as autoimmune conditions, inflammatory bowel diseases, chronic lung and kidney diseases and Diabetes mellitus [7], we analyzed concomitant medical comorbidities to evaluate a potential influence of SD on vaccination rates. Interestingly, patients with SD were, for the most part, even less likely to be vaccinated than those without SD. Even in the group with autoimmune conditions, known to be especially prone to HZ reactivation [15], no patient was vaccinated. A potential confounder on vaccination rates by SD in our study population of hematological and oncological patients is therefore highly unlikely.

### Limitations

Study participants were recruited in four large outpatient clinics in Berlin, Germany, therefore numbers are prone to regional and institutional factors. Nonetheless, clinics are independently organized and located in different neighborhoods with different socio-economic status, representing a broad spectrum of patients and treating physicians. Additionally, results of a population based study for Germany confirm a Shingrix^®^ vaccination rate for the “population with relevant diseases” of 1.1–3.5%, suggesting our real-life data to be realistic [16].

As data were collected during the COVID-19 pandemic, access to health facilities was limited, potentially reducing opportunities to vaccinate against HZ. In addition, priority of treatment and vaccinations focused on oncological treatment essentials and COVID-19 related modalities. Although official recommendations in the pandemic stressed the need for other vaccination besides targeting COVID-19 [17], this might have affected vaccination rates for Shingrix^®^. A general caution to use a new vaccine with not fully established adverse events for hematological and oncological patients might also have reduced uptake. Furthermore, availability of Shingrix^®^ was partly limited in 2020 due to insufficient production capacities to meet unexpected high demand worldwide. Still, as outlined, we were able to increase vaccination rates by an intervention, proving the potential for higher rates, even under these conditions [10].

## 5. Conclusions

Vaccination rates for Shingrix^®^ are still unsatisfactorily low in the high-risk population of hematological and oncological patients. There is a pressing need to address this situation and overcome barriers.

## Figures and Tables

**Table 1 healthcare-10-01524-t001:** Shingrix^®^ vaccination rates according to underlying cancer disease in frequency (n) and percentage (%).

**Hematological Cancer (N = 78)** **n (%)**							
6 (7.7)							
Acute Leukemia (N = 3)	Chronic Leukemia (N = 1)		Lymphoma (N = 25)		Multiple Myeloma (N = 43)		Other (N = 6)
0 (0.0%)	0 (0.0%)		1 (4.0%)		5 (11.6)		0 (0.0%)
**Solid Cancer (N = 292)** **n (%)**							
5 (1.7)							
Endocrine Carcinoma (N = 1)	Urologic Carcinoma (N = 26)	Gynecol. Carcinoma (N = 41)	Gastrointest. Carcinoma (N = 80)	Skin Carcinoma (N = 9)	Thoracic Carcinoma (N = 116)	Throat Carcinoma (N = 13)	Other (N = 6)
0 (0.0%)	2 (7.7)	1 (2.4)	0 (0.0%)	0 (0.0%)	2 (1.7)	0 (0.0%)	0 (0.0%)
**Total (N = 370)** **n (%)**							
11 (3.0)							

**Table 2 healthcare-10-01524-t002:** Shingrix^®^ vaccination rates in oncological patients according to secondary diagnoses (SD) in frequency (n/N) and percentage (%).

Secondary Diagnosis	SD Not Documented n/N (%)	SD Documented n/N (%)
Cardiac	5/154 (3.2)	6/216 (2.8)
Pulmonary	2/289 (3.5)	1/80 (1.2)
Nephrological	10/327 (3.1)	1/42 (2.3)
Vascular	11/329 (3.3)	0/41 (0.0)
Gastrointestinal	9/298 (3.0)	2/71 (2.8)
Hepatic	10/346 (2.9)	1/23 (4.3)
Endocrine	10/269 (3.7)	1/100 (1.0)
Autoimmune	11/353 (3.1)	0/17 (0.0)

## Data Availability

Data are stored in the IT system of the Vivantes Hospital Neukoelln and available by request.

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
