# Peer review of "Herpes Zoster Vaccination Rates in Hematological and Oncological Patients—Stock Taking 2 Years after Market Approval"

_healthcare, 2022, doi:10.3390/healthcare10081524_

Round 1

Reviewer 1 Report

The authors should be congratulated for their manuscript entitled: "Herpes zoster vaccination rates in haematological ..."

The data is clearly presented and analyzed.

The authors should research, present and discuss that in the first year (2 years?) after approval there was such high demand for Shingrix worldwide and in Germany, that the demand could not be met by the manufacturer for all patients Germany. Could this have contributed to the low rate of HZ vaccination uptake reported in the current study?

Author Response

Dear reviewer,

thank you for your time and effort and the positive review.

We made a spelling check for English.

Point 1 - The authors should research, present and discuss that in the first year (2 years?) after approval there was such high demand for Shingrix worldwide and in Germany, that the demand could not be met by the manufacturer for all patients Germany. Could this have contributed to the low rate of HZ vaccination uptake reported in the current study?

Resonse 1 - We included the issue of high demand in the Limitation section, as you correctly suggested: “Furthermore, availability of Shingrix® was partly limited in 2020 due to insufficient production capacities to meet unexpected high demand worldwide.

Thank you again for our input.

Reviewer 2 Report

The manuscript describes data extracted from the 'Easy Vaccination in Oncology (EVO) Study" (https://www.aerzteblatt.de/int/archive/article/225962) to investigate the uptake of vaccination against herpes zoster in hematological and oncological.  As the EVO Study citation is not available in English it's not feasible to determine how the data are derived for the current study.   Although the topic of the manuscript is of significant importance to understanding the role of vaccination against herpes zoster in hematological and oncological patients, the design of the current study seems somewhat obscure.  Moreover, this manuscript does not describe a study, but an overview of data reported in the EVO Study, and it seems unusual that data would be lifted directly from another study reporting on vaccination against infectious agents including varicella-zoster virus, the causative agent of herpes zoster.

An important clarification is that Shingrix is not an inactivated vaccine.  It's a subunit vaccine based on VZV glycoprotein E (gE), which is an essential protein required for replication.  This particular vaccine uses an aggressive adjuvant that can cause considerable reaction at the injection site.  With a limited efficacy of 64% in immunocompromised patients compared to 97% in healthy individuals (see lines 117-119) coupled with a high propensity of Shingrix vaccine to cause adverse events (https://shingrixhcp.com/efficacy-safety/adverse-reactions/), this might be a reason why the current uptake in hematological and oncological patients has been limited. Both patients or doctors might not want to add to the burden of already difficult treatments.  This should be addressed in the limitations section.

Author Response

Dear reviewer,

thank you for your time and effort invested.

We revised the text regarding English language and style.

Point 1 - The manuscript describes data extracted from the 'Easy Vaccination in Oncology (EVO) Study" (https://www.aerzteblatt.de/int/archive/article/225962) to investigate the uptake of vaccination against herpes zoster in hematological and oncological.  As the EVO Study citation is not available in English it's not feasible to determine how the data are derived for the current study. Although the topic of the manuscript is of significant importance to understanding the role of vaccination against herpes zoster in hematological and oncological patients, the design of the current study seems somewhat obscure.  Moreover, this manuscript does not describe a study, but an overview of data reported in the EVO Study, and it seems unusual that data would be lifted directly from another study reporting on vaccination against infectious agents including varicella-zoster virus, the causative agent of herpes zoster.

Response 1 - Regarding the availability of an English version of the original publication it is expected to be online the following days, as the English full text will be published approximately two weeks after the German print edition has been published. In short: Data was originally raised in the context of an interventional study (EVO) trying to show efficacy of an intervention to increase vaccination rates in the described patient group. To present first data specifically for Herpes/Varicella zoster vaccination rates in this patient group we extracted and analyzed only patients not affected by the intervention (baseline and non-intervention follow-up), equivalent to a cross-sectional study. Herpes zoster data presented in the EVO publication were pooled differently and not analyzed and discussed in detail.

Point 2 - An important clarification is that Shingrix is not an inactivated vaccine.  It's a subunit vaccine based on VZV glycoprotein E (gE), which is an essential protein required for replication.  

Response 2 - Thank you for your valid remark regarding the vaccine itself. We changed the denomination of the vaccine.

Point 3 - This particular vaccine uses an aggressive adjuvant that can cause considerable reaction at the injection site.  With a limited efficacy of 64% in immunocompromised patients compared to 97% in healthy individuals (see lines 117-119) coupled with a high propensity of Shingrix vaccine to cause adverse events (https://shingrixhcp.com/efficacy-safety/adverse-reactions/), this might be a reason why the current uptake in hematological and oncological patients has been limited. Both patients or doctors might not want to add to the burden of already difficult treatments.  This should be addressed in the limitations section.

Response 3 - We added a sentence in the Limitation section: “A general caution to use a new vaccine with not fully established adverse events for hematological and oncological patients might also have reduced uptake.” It has to be said our experience does not show specifically high AEs for Shingrix® vaccination and available numbers do not notably differ from other vaccines. So we feel more comfortable to keep a general statement in this regard.

Thank you again for you input.

Reviewer 3 Report

  • Kiderlen et al. present a manuscript on Herpes Zoster vaccination rates among oncological patients. They based their analysis on data of the Easy Vaccination in Oncology (EVO) Study, which was an interventional study aiming to increase vaccination rates among oncological patients for a specific subset of vaccines recommended in that setting. The authors present results showing vaccination of only about 3% of the studied population and conclude that factors such as: short time of availability on the market, lack of awarness of the patients, limited availability and access to vaccination due to COVID-19 pandemic could have contributed to the low vaccination rate. The manuscript is written clearly and addresses an important and relevant issue for the oncological patients. The results of this study, confirm the results of a previous population-based study with regard to rate of vaccination.
  • The main weakness of the manuscript is a complete lack of statistical analysis. It would be important to include statistical analysis for all of the data, but particularly for: data showing more frequent vaccination with Shingrix for patients with hematological malignancies in comparison to patients with solid tumors and higher rates of vaccination of patients with multiple myeloma among the hematological malignanciy patients.
  • Specific comments:
  • line 83: under gastrointestinal SD does the chronic refer to colitis or is there perhaps a word missing
  • table 1 - there is a problem with formatting - the n (%) of solid cancer patients is shifted to the left/not centered

Author Response

Dear reviewer,

thank you for our valuable time and effort invested.

Point 1 - Line 83: under gastrointestinal SD does the chronic refer to colitis or is there perhaps a word missing. Table 1 - there is a problem with formatting - the n (%) of solid cancer patients is shifted to the left/not centered.

Response 1 - our valid remarks ragarding fomating and missing words were addressed.

Point 2 - The main weakness of the manuscript is a complete lack of statistical analysis. It would be important to include statistical analysis for all of the data, but particularly for: data showing more frequent vaccination with Shingrix for patients with hematological malignancies in comparison to patients with solid tumors and higher rates of vaccination of patients with multiple myeloma among the hematological malignanciy patients.

Response 2 - Regarding the statistics allow us to respond. We considered further statistical analysis but decided against it. One reason was numbers are clearly too low to show statistical differences, but most importantly it was not our aim to present significant differneces between the groups but provide first-time prevalence data.

I hope we addressed your comments and send our kind regards.

Round 2

Reviewer 2 Report

There are a good number of grammatical errors that need addressing; too many to list.

Author Response

Thank you.

We did proof reading with a native speaker.